# A Security Transmission and Storage Solution about Sensing Image for Blockchain in the Internet of Things

**DOI:** 10.3390/s20030916

**Published:** 2020-02-09

**Authors:** Yunfa Li, Yifei Tu, Jiawa Lu, Yunchao Wang

**Affiliations:** 1Key Laboratory of Complex Systems Modeling and Simulation, School of Computer Science and Technology, Hangzhou Dianzi University, Hangzhou 310018, China; yifeitu@hdu.edu.cn (Y.T.); wangyunchao0107@icloud.com (Y.W.); 2Department of Mechanical, Materials and Manufacturing Engineering, Faculty of Science and Engineering, University of Nottingham Ningbo China, Ningbo 315100, China; Jiawa.Lu@nottingham.edu.cn

**Keywords:** Internet of things, blockchain, transmission, storage, security

## Abstract

With the rapid development of the Internet of Things (IoT), the number of IoT devices has increased exponentially. Therefore, we have put forward higher security requirements for the management, transmission, and storage of massive IoT data. However, during the transmission process of IoT data, security issues, such as data theft and forgery, are prone to occur. In addition, most existing data storage solutions are centralized, i.e., data are stored and maintained by a centralized server. Once the server is maliciously attacked, the security of IoT data will be greatly threatened. In view of the above-mentioned security issues, a security transmission and storage solution is proposed about sensing image for blockchain in the IoT. Firstly, this solution intelligently senses user image information, and divides these sensed data into intelligent blocks. Secondly, different blocks of data are encrypted and transmitted securely through intelligent encryption algorithms. Finally, signature verification and storage are performed through an intelligent verification algorithm. Compared with the traditional IoT data transmission and centralized storage solution, our solution combines the IoT with the blockchain, making use of the advantages of blockchain decentralization, high reliability, and low cost to transfer and store users image information securely. Security analysis proves that the solution can resist theft attacks and ensure the security of user image information during transmission and storage.

## 1. Introduction

Originated in the 1990s, the concept of the Internet of Things (IoT) is to build on the Internet and extend the Internet. The Internet is the interaction between people through the network, and the IoT is the information exchange between people, between people and things, and between things. The basic feature of the IoT is the intelligent sensing collection of data, which are then reliably transmitted to the server for calculation and intelligent processing. The IoT has experienced rapid development for more than two decades, and the number of IoT devices has grown exponentially, a trend that greatly improves social productivity and efficiency as well as makes people’s lives more convenient and intelligent. Today, the IoT has been used in every corner of people’s lives, including smartphones, cars, smart homes, and other mobile devices. With the official arrival of the 5G era, experts estimate that, by 2020, there will be more than 30 billion IoT devices worldwide. However, these many IoT devices will generate many data, which are transmitted through wireless networks. It is easy to cause security attacks, such as theft and forgery of malicious attackers. Therefore, to securely transmit massive IoT data is one of the important issues in the development of the IoT.

The issue of secure storage of IoT data is also a key concern in the field of IoT. Currently, many IoT data storage solutions store data on traditional centralized servers. However, centralized data storage has the following disadvantages. Firstly, all data are stored centrally on a main server, s thuso the performance requirements, environmental requirements, and maintenance costs for the main server are extremely high. The server must have a very high storage capacity to store many IoT data. Once the server fails, the entire IoT data storage system will not work properly. Secondly, because all IoT devices are connected to the centralized server, the connection will cause network congestion, high network delay, and inefficient data storage of the IoT. Finally, once the centralized server is attacked by malicious attackers, the security of IoT data will be greatly threatened.

As a new application mode of distributed data storage, the blockchain can well solve the problems that easily occur in centralized data storage. Originated in 2008, the essence of blockchain, which is the underlying technology for the development of Bitcoin, is to centrally maintain a reliable database solution through decentralization and detrust. The blockchain divides the data into a series of data blocks. Each data block can verify the validity of the data and generate the next data block, which is connected in a chronological order into a chain data structure. Blockchain storage has the following advantages. (1) Decentralization: There is no centralized hardware in blockchain storage. Using point-to-point technology, the power and obligations of each node are equal, thus the destruction of a node does not affect the normal work of the entire system. (2) Detrust: Since the interaction between nodes passes a fixed algorithm, there is no need to trust each other in the form of public identity. At the same time, the more nodes there are, the stronger is the security of the system. (3) Low cost: Blockchain storage has good data deduplication ability to reduce data redundancy and costs. (4) Information cannot be tampered: When data information is added to the blockchain through verification, the information cannot be tampered, a fact that effectively prevents forgery attacks on data by attackers and ensures the reliability of data storage. (5) High transparency: Except for individual private information being encrypted, the entire data storage system is transparent and open, so that nodes can maintain the system together.

To solve the security problems of the IoT data during the transmission and storage process, this article proposes a secure transmission and storage solution for the sensing images facing the blockchain in the IoT. It combines blockchain technology with the IoT, and divides the IoT data into different blocks for signature verification and storage. Our contributions are as follows:

(1) We propose a block division algorithm for image sensing and data sensing of the smart image sensor. By judging the number of distributed storage servers in the blockchain and the storage status, the number of divided blocks of user image sensing data is determined and divided.

(2) We propose a public key generation algorithm for the sensing data blocks. The public key is calculated by an intelligent algorithm with *n* numbers randomly generated by the intelligent image sensor, and all storage servers in the blockchain use the same public key.

(3) We propose a private key generation and update algorithm for the sensing data blocks. The private key of each sensing data block in different periods in the cycle is different, and the private key of the sensing data block in one time period can generate the private key of the data block in the next time period through an update algorithm.

(4) We propose a signature and verification algorithm for the sensing data blocks. Each data block is signed by a different private key and sent to the storage server through a signature algorithm. Each storage server uses the public key to authenticate and store the signature information of the sent data blocks.

The organization of this article is as follows: Section 2 introduces the related work of the IoT data storage combined with blockchain. Section 3 introduces the specific scheme of security transmission and storage solution about sensing image for blockchain in the IoT. The security of the scheme is analyzed and proved in Section 4. The last section presents conclusions and future work.

## 2. Related Work

In the recent years, people have done a lot of research in the direction of secure transmission and storage of IoT data combined with blockchain technology, and have made some achievements.

Blockchain technology is used in many fields, such as smart grid, industry, and medical fields. Gai et al. [1] proposed the Privacy-enabled Blockchain-enabled Transaction (PBT) model to solve the privacy problem of energy transaction users in the smart grid. This solution first proposes a noise-based privacy protection method, which does not need to screen individual privacy, and hides the transaction distribution trend of adjacent energy trading systems with blockchain. Secondly, they designed a privacy protection mechanism to achieve the effect of the differential privacy algorithm, which introduces virtual accounts and divided accounts to change the function of transaction records without reducing other performance such as accuracy. Liang et al. [2] proposed a blockchain-based secure data transmission scheme based on an optimized FaBric architecture, which was applied to the Industrial IoT to solve the security problem in a blockchain-based power network. This solution is based on the blockchain’s dynamic key sharing mechanism, and implements network consensus and identity verification by dynamically adding power nodes. Through security experiment analysis, this scheme reduces the communication load of node consensus, simplifies the consensus flow in data transmission, improves the security of data transmission, and reduces overhead. Shen et al. [3] proposed a blockchain-based medical encrypted image retrieval scheme to protect user image privacy in the medical IoT environment. Firstly, this solution uploads images to the hospital server through IoT devices. The hospital server extracts different types of medical image features, and uses the image features to be encrypted and stored by Secure Multi-party Computation (SMC). Secondly, the image retrieval service indexes all encrypted images. When a user makes a retrieval request, the service initiates a smart contract to send a request for the user, and feeds back the retrieval results based on the image similarity.

The combination of blockchain technology and edge computing has huge potential and has gradually become a disruptive technology that promotes social progress. Li et al. [4] proposed a security scheme for the storage and protection of IoT data based on blockchain. The solution uses edge computing to perform data calculations for IoT devices and forwards the data to storage. In addition, the solution uses certificateless encryption technology to establish a convenient identity verification system for blockchain-based IoT applications, and overcomes the shortcomings of certificate encryption technology by broadcasting the public key of the IoT device. Gai et al. [5] combined the blockchain and edge computing technology, and proposed a model permission blockchain edge model (PBEM-SGN) suitable for smart grids to solve two important issues in smart grids, namely privacy protection and energy security. The transparent operation in this model helps to detect incorrect energy usage behaviors, thereby reducing or avoiding energy-related attacks to control and map SGN network nodes. Guo et al. [6] proposed a distributed trusted authentication system based on blockchain and edge computing. The system consists of a physical network layer, a blockchain edge layer, and a blockchain network layer. First, they designed an optimized practical Byzantine fault-tolerant consensus algorithm to build a consortium blockchain for storing authentication data and logs. It guarantees trusted authentication and enables traceability of terminal activities. In addition, edge computing is applied to blockchain edge nodes to provide name resolution and edge authentication services based on smart contracts. Secondly, the system uses asymmetric encryption technology to prevent the connection between the node and the terminal from being attacked. At the same time, the system proposes a cache strategy based on edge computing to improve the hit rate. Pan et al. [7] designed and prototyped an edge IoT framework based on blockchain and smart contracts. The core idea is to integrate the licensed blockchain with an internal currency or system to link the edge cloud resource pool with each IoT device’s linked account and resource usage, as well as the behavior of the IoT device. EdgeChain uses a credit-based resource management system to control how many resources IoT devices get from the edge server based on predefined rules of priority, application type, and past behavior. Smart contracts are used to enforce rules and policies and regulate the behavior of IoT devices in an undeniable, automated way. All IoT activities and transactions are recorded in the blockchain for secure data logging and auditing.

Reducing resource consumption is one of the advantages of the combination of blockchain technology and the IoT, especially for the limited resources of IoT devices. Kim et al. [8] optimized the Byzantine Fault Tolerance (BFT) consensus algorithm for the lightweight IoT network based on blockchain, and proposed a storage compression consensus (SCC) algorithm. The algorithm compresses the blockchain in each device to ensure storage capacity. When the lightweight device does not have enough storage space, it uses the SCC algorithm to compress the blockchain. Results of simulation and comparative experiment analysis show that the algorithm has a more obvious compression effect on the storage capacity of the blockchain, and can be used to form a storage-efficient lightweight IoT network. Shahid et al. [9] proposed a lightweight and scalable blockchain framework for resource-constrained IoT sensor devices and named it “sensor chai”. This framework realizes a lightweight blockchain through three steps. Firstly, the global blockchain is divided into smaller disjoint local blockchains in the spatial domain to ensure that the storage space of the local blockchain is always smaller than that of the traditional blockchain. Secondly, the lifecycle of the local blockchain is limited to control its size in the time domain. Finally, a sensor node can only retain up to one local blockchain. Experimental analysis shows that the framework can significantly reduce resource consumption and can be expanded as the network scale increases, while retaining key information of the IoT system. To solve the Proof of Work (PoW) computation problem when the resources of the IoT device are limited, Doku et al. [10] proposed a scheme that combines blockchain and the IoT and mines a network with limited node resources. The comparative experimental analysis shows that the scheme has stronger scalability and more efficient use of resources.

In the IoT, the identity authentication of nodes is required to ensure the security and reliability of data transmission, and the combination of blockchain technology can eliminate the need for trusted third parties to participate in authentication. Almadhoun et al. [11] proposed a user authentication scheme using blockchain atomized nodes, combined with Ethereum smart contracts to implement blockchain-based large-scale IoT device authentication without the need for an intermediate third party. The fog node is used to reduce the processing load of the IoT device during the authentication process and reduce the overhead of connecting to the Ethereum blockchain network interface. The security analysis shows that the scheme can resist known eavesdropping, replay, and DoS attacks and has good security. Manzoor et al. [12] proposed a blockchain-based proxy re-encryption scheme in the context of the IoT. This solution stores IoT data in a distributed cloud, and uses dynamic smart contracts to connect sensors to share IoT data with data users without the participation of trusted third parties. In addition, the scheme uses an effective proxy re-encryption algorithm, allowing only the owner and those present in the smart contract to see the data. Wang et al. [13] proposed the Blockchain and Bilinear mapping based Data Integrity Scheme (BB-DIS) for large-scale IoT data. This solution cuts IoT data into fragments and generates Homomorphic Verifiable Tags (HVT) for sample verification. In addition, this solution achieves data integrity based on the characteristics of bilinear mapping in the form of blockchain transactions. The analysis of experimental results shows that the BB-DIS scheme significantly improves the efficiency of large-scale IoT data integrity verification, and does not require the trust of Third Party Auditors (TPA).

Due to the decentralized nature of traditional blockchain technology, nodes are equal. Rashid et al. [14] proposed a multi-layer secure network model of the IoT network based on blockchain technology. This model solves the problems related to the actual deployment of blockchain technology by dividing the IoT network into a multi-layer decentralized system. The model uses Genetic Algorithms (GA) and Particle Swarm Optimization (PSO) to divide the network into *K* unknown clusters, and uses a local authentication mechanism in the cluster handed over by each Cluster Head (CH). This model takes advantage of the high security and credibility of the blockchain technology, and provides an authentication mechanism that enables the CHs to communicate with each other and with the base station without central authorization. Ayoade et al. [15] proposed a decentralized data management system for blockchain-based IoT devices. The system uses a blockchain platform to provide the decentralized IoT data access management, and uses smart contracts to provide the equal data access management privileges between IoT users and IoT service providers. At the same time, the system uses a trusted execution environment (Intel SGX) to provide data storage for secure data storage. In addition, the system uses Ethereum smart contracts to provide a complete system implementation on a real blockchain platform.

Different from the traditional blockchain technology, some scholars have introduced special nodes in the blockchain, such that the nodes are no longer equal, to achieve some centralized control functions. Zhu et al. [16] proposed a Controllable Blockchain Data Management (CBDM) model, which introduced a specific node. This particular node is also called a Trust Authorization Authority (TA) node, and it has a higher level of voting authorization than other participant nodes. The system is configured with a veto power, which is specifically used to prevent malicious voting. In addition, our model utilizes cloud storage to achieve storage efficiency and the data in each block store only metadata to improve block building efficiency and minimize distributed storage waste. Zhou et al. [17] proposed a Threshold Secure Multi-Party Computing (TSMPC) protocol and combined it with the blockchain. Afterwards, they proposed a blockchain-based threshold IoT service system named BeeKeeper. In the BeeKeeper system, the server processes user data and generates responses by performing homomorphic calculations on the information, but the server cannot get any user privacy information from the information. Moreover, the system can verify the data and response to detect malicious nodes. In addition, any node can be a leader’s server. Based on performance evaluation and analysis, the system has good efficiency and fault tolerance.

Considering the above related work, due to the advantages of blockchain technology, being decentralization, high reliability, and low cost, the data storage solution combining the IoT and the blockchain is better than the traditional IoT data transmission and centralized storage solutions in terms of security, performance, and resource utilization. Based on the above advantages, a security transmission and storage solution is proposed to sense image for blockchain in the IoT. This solution can effectively resist the forgery and theft attacks of image information by attackers and ensure that the transmission and storage of user image information is more secure.

## 3. Security Transmission and Storage Solution of Induction Images

This section introduces a specific solution for the security transmission and storage solution about sensing image for blockchain in the Internet of Things. This solution mainly contains five algorithms: (1) blocking algorithm for image sensing and data sensing of the smart image sensor; (2) public key generation algorithm for the sensing data blocks; (3) private key generation and update algorithm for the sensing data blocks; (4) signature algorithm for the sensing data blocks; and (5) signature verification algorithm for the sensing data blocks. Through these five algorithms, secure transmission and storage of blockchain-oriented sensor information can be achieved in the IoT. This solution uses intelligent image sensors and storage servers. The intelligent image sensor is a device with a camera and computing power that can collect user’s image information for analysis. In this solution, the block division, the generation of public and private keys, and the update and encryption of private keys are all implemented through the intelligent image sensor. The storage server can store user image information divided into blocks. In this solution, the authentication function is implemented by the storage server. The symbols and definitions used in this article are shown in Table 1.

### 3.1. Implementation Process

The schematic diagram of the realization process of the secure transmission and storage solution about sensing image for blockchain in the IoT is shown in Figure 1 and described as follows:

Step 1: The intelligent image sensor perceives the user image based on the surrounding scene using the image sensing of the intelligent image sensor and the block division algorithm of the sensing data.

Step 2: The intelligent image sensor divides the perceived user image using the image sensing and block data partitioning algorithm of the sensing data.

Step 3: The smart image sensor uses the public key generation algorithm of the sensing data block to generate a public key according to the status of the divided sensing data block, and securely transmits this public key to each storage server in the blockchain.

Step 4: The storage server sends a confirmation message to the smart image sensor after receiving the public key sent by the smart image sensor.

Step 5: After receiving the confirmation message from the storage server, the smart image sensor uses the signature algorithm of the sensing data block to sign the sensing data block with different private keys in different time periods and sends the signed message to the corresponding storage server.

Step 6: After receiving the sensing data blocks transmitted by the intelligent image sensor in different time periods, the storage server uses the signature verification algorithm of the sensing data blocks to quickly perform signature verification on different sensing data.

Step 7: If the signature verification is passed, the storage server receives the sensing data block and saves the received data block.

Step 8: End.

### 3.2. Algorithm 1: Blocking Algorithm for Image Sensing and Data Sensing of the Smart Image Sensor

Step 1: The intelligent image sensor judges whether there are user images that need to be collected according to the surrounding scenes. If so, go to Step 3; otherwise, go to Step 2.

Step 2: The smart image sensor waits for three seconds before going to Step 1.

Step 3: The intelligent image sensor collects user images P(u) according to the surrounding scene.

Step 4: The intelligent image sensor determines the number n of the sensing data of the user image according to the number of distributed storage servers and storage status in the blockchain.

Step 5: The intelligent image sensor determines the division mode of the image sensing data block according to the storage status of the distributed storage server in the blockchain and divides it, that is, $SP_{i}\left(u\right)\neq \Phi$, $SP_{i}\left(u\right)\subset P\left(u\right)$, $SP_{i}\left(u\right)\cap SP_{j}\left(u\right)=\Phi$, $P\left(u\right)=\overset{n}{\underset{i=1}{⋃}}SP_{i}\left(u\right)$.

Step 6: The intelligent image sensor determines whether the sensing data of the user image has been divided. If so, go to Step 7; otherwise, go to Step 5.

Step 7: End.

### 3.3. Algorithm 2: Public Key Generation Algorithm for the Sensing Data Blocks

Step 1: The intelligent image sensor randomly selects two large prime numbers *P* and *Q* and calculates the product of *P* and *Q*, that is, N=P∗Q.

Step 2: The intelligent image sensor calculates ϕ(N)=(P−1)∗(Q−1) and randomly selects a generator $R_{1}$ of $Z_{N}^{*}$ of module *N*, that is, $R_{1}=rand\left(Z_{N}^{*}\right)$.

Step 3: The intelligent image sensor randomly selects a relatively large integer *L*, calculates $A=L^{2}$, and makes the value of *A* greater than the number of divided blocks *n* of the sensing data of the user image, that is, A>n.

Step 4: The smart image sensor randomly selects an integer $a_{i}$ in interval [A,2A) (i.e.,, $a_{i}\in \left[A,2A\right)$, i∈{1,2,3,…,n}, n<A), makes $a_{i}$ and ϕ(N) coprime, and makes $a_{i}$ and $a_{j}$ coprime (i.e.,, $\gcd(a_{i},\phi \left(N\right))=1$, $\gcd(a_{i},a_{j})=1$, i∈{1,2,3,…,n}, j∈{1,2,3,…,n}, i≠j).

Step 5: At the beginning of the signature calculation, the smart image sensor sets the initial time to 0, selects a relatively large cycle period *T* according to the system setting requirements (*T* is an integer, and the cycle period *T* is greater than or equal to the system each time maximum cycle working time), and calculates $B_{1}=A\left(1+\left(i-1\right)/T\right)$, $B_{2}=A\left(1+i/T\right)$ (1≤i≤T).

Step 6: The smart image sensor randomly selects an integer $b_{i}$ in interval $\left[B_{1},B_{2}\right)$ (i.e.,, $b_{i}\in \left[B_{1},B_{2}\right)$, i∈{1,2,3,…,k}, k≤T), makes $b_{i}$ and ϕ(N) coprime, and makes $b_{i}$ and $b_{j}$ coprime (i.e.,, $\gcd(b_{i},\phi \left(N\right))=1$, $\gcd(b_{i},b_{j})=1$, i∈{1,2,3,…,k}, j∈{1,2,3,…,k}, i≠j).

Step 7: The intelligent image sensor calculates $MA=a_{1}*a_{2}*\ldots *a_{n}=\prod_{i=1}^{n}a_{i}$.

Step 8: In the cycle *T*, the intelligent image sensor calculates the key parameter $t_{i,1}={R_{1}}^{MA/a_{i}}\;\mathrm{mod}\;N$ (1≤i≤n) of each sensing data block after the sensing data are segmented.

Step 9: The smart image sensor calculates $PK\left(P\left(u\right)\right)={R_{1}}^{MA*\prod_{m=1}^{T}b_{m}}\;\mathrm{mod}\;N$, and uses PK(P(u)) as the public key of the user’s image sensing data P(u) (i.e.,, the public key of each storage server in the blockchain to authenticate the signature information of the sent data block).

Step 10: The smart image sensor determines whether the public key of the user’s image sensing data P(u) has been generated. If not, go to the Step 1; otherwise, go to the Step 11.

Step 11: The smart image sensor saves the parameter information $\xi =\{t_{1,1}||t_{2,1}||t_{3,1}||\ldots ||t_{i,1}||\ldots ||t_{n,1}\}$, $\psi =\{b_{1}||b_{2}||\ldots ||b_{j}||\ldots ||b_{T}\}$ and *N* during the calculation of the public key, and sends the generated public key PK(P(u)) of the user image sensing data to each storage server in the blockchain through the network.

Step 12: End.

### 3.4. Algorithm 3: Private Key Generation and Update Algorithm for the Sensing Data Blocks

Step 1: The intelligent image sensor calculates $S_{i,1}\left(SP_{i}\left(u\right)\right)={t_{i,1}}^{b_{2}•b_{3}•b_{4}\ldots b_{T}}\;\mathrm{mod}\;N$ (i∈{1,2,3,…,n}), $t_{i,2}={t_{i,1}}^{b_{1}}\;\mathrm{mod}\;N$ based on the parameter information $\xi =\{t_{1,1}||t_{2,1}||t_{3,1}||\ldots ||t_{i,1}||\ldots ||t_{n,1}\}$, $\psi =\{b_{1}||b_{2}||\ldots ||b_{j}||\ldots ||b_{T}\}$, and *N* in the public key calculation process.

Step 2: The intelligent image sensor calculates the private key $SK_{i,1}\left(SP_{j}\left(u\right)\right)=\left\{i||1|\right|\alpha_{i}\left|\right|b_{1}\left|\right|S_{i,1}\left(SP_{i}\left(u\right)\right)\left|\right|t_{i,2}\left||\psi \right\}$ of the sensing data block $SP_{i}\left(u\right)$ during the period t=1 of the cycle *T* (t⊂T).

Step 3: In the period t=m (m∈{1,2,3,…,k−1}, k≤T) within the cycle *T*, the smart image sensor generates a private key $SK_{i,m}\left(SP_{i}\left(u\right)\right)=\left\{i||m|\right|\alpha_{i}\left|\right|b_{m}\left|\right|S_{i,m}\left(SP_{i}\left(u\right)\right)\left|\right|t_{i,m+1}\left||\psi \right\}$ ($S_{i,m}\left(SP_{i}\left(u\right)\right)={t_{i,1}}^{b_{j}•b_{m+1}•b_{m+2}\ldots b_{T}}\;\mathrm{mod}\;N$, $t_{i,m+1}={t_{i,m}}^{b_{m}}\;\mathrm{mod}\;N$) for the sensing data block $SP_{i}\left(u\right)$ (i∈{1,2,3,…,n}).

Step 4: During the period t=m+1 (m+1∈{1,2,3,…,k}) within the cycle *T*, the intelligent image sensor generates a private key $SK_{i,m+11}\left(SP_{i}\left(u\right)\right)=\{i||m+1||\alpha_{i}\left|\right|b_{m+1}\left|\right|S_{i,m+1}\left(SP_{i}\left(u\right)\right)\left|\right|t_{i,m+2}\left||\psi \right\}$ ($S_{i,m+1}\left(SP_{i}\left(u\right)\right)={t_{i,1}}^{b_{j}•b_{m}•b_{m+2}\ldots b_{T}}\;\mathrm{mod}\;N$, $t_{i,m+2}={t_{i,m+1}}^{b_{m+1}}\;\mathrm{mod}\;N$) for the sensing data block $SP_{i}\left(u\right)$ (i∈{1,2,3,…,n}).

Step 5: End.

### 3.5. Algorithm 4: Signature Algorithm for the Sensing Data Blocks

In the user image P(u), it is assumed that the smart image sensor needs to transmit the divided sensing data block $SP_{i}\left(u\right)$ to a storage server in the blockchain. After receiving the confirmation message from this storage server, the smart image sensor block $SP_{i}\left(u\right)$ performs the following operations.

Step 1: In the period t=m (m∈{1,2,3,…,k}, k≤T) within the cycle *T*, the intelligent image sensor randomly selects a generator $R_{2}$ of $Z_{N}^{*}$ of module *N* for the sensing data block $SP_{i}\left(u\right)$ (i∈{1,2,3,…,n}), that is, $R_{2}=rand\left(Z_{N}^{*}\right)$.

Step 2: The intelligent image sensor calculates $x={R_{2}}^{a_{i}•b_{m}}\;\mathrm{mod}\;N$ for the sensing data block $SP_{i}\left(u\right)$ (i∈{1,2,3,…,n}).

Step 3: The intelligent image sensor calculates $\varpi =H(i||m||\alpha_{i}||b_{m}||x||SP_{i}\left(u\right))$, $y=(R_{2}•S_{i,m}{\left(SP_{i}\left(u\right)\right)}^{\varpi })\;\mathrm{mod}\;N$ for the sensing data block $SP_{i}\left(u\right)$ (i∈{1,2,3,…,n}).

Step 4: The smart image sensor signs the sensing data block $SP_{i}\left(u\right)$ (i∈{1,2,3,…,n}), that is, $\eta =sign_{SK_{i,m}\left(SP_{i}\left(u\right)\right)}\left(SP_{i}\left(u\right)\right)\leftarrow \left(i||m|\right|\alpha_{i}\left|\right|b_{m}\left||\varpi ||y\right)$.

Step 5: The smart image sensor generates a timestamp value $TS_{i}$ of the smart image sensor, and transmits the signature message $\eta =sign_{SK_{i,m}\left(SP_{i}\left(u\right)\right)}\left(SP_{i}\left(u\right)\right)$ and the timestamp value $TS_{i}$ of the sensing data block $SP_{i}\left(u\right)$ (i∈{1,2,3,…,n}) to the corresponding storage server (Storage Server l) in the blockchain.

Step 6: End.

### 3.6. Algorithm 5: Signature Verification Algorithm for the Sensing Data Blocks

Step 1: The corresponding storage server $STS_{l}$ in the blockchain receives the signed message $\eta =sign_{SK_{i,m}\left(SP_{i}\left(u\right)\right)}\left(SP_{i}\left(u\right)\right)$ (i∈{1,2,3,…,n}, n<A) and the timestamp value $TS_{i}$ sent by the smart image sensor.

Step 2: Storage server $STS_{l}$ generates a storage server timestamp $TSTS_{i}$. First, check whether the session delay $TSTS_{i}-TS_{i}$ is within the allowable time interval Δt. If $TSTS_{i}-TS_{i}\geq \Delta t$, the session times out, and go to Step 1. Then, the storage server $STS_{l}$ uses signature message $\eta =sign_{SK_{i,m}\left(SP_{i}\left(u\right)\right)}\left(SP_{i}\left(u\right)\right)$ to calculate $f=y\;\mathrm{mod}\;N=(\left(R_{2}•S_{i,m}{\left(SP_{i}\left(u\right)\right)}^{\varpi }\right)\;\mathrm{mod}\;N)\;\mathrm{mod}\;N$, and determines whether the following equation f=0modN holds. If it is true, go to Step 3; otherwise, go to Step 1.

Step 3: The storage server $STS_{l}$ uses the public key $PK\left(P\left(u\right)\right)={R_{1}}^{MA*\prod_{m=1}^{T}b_{m}}\;\mathrm{mod}\;N$ of the user image sensing data and the signature message $\eta =sign_{SK_{i,m}\left(SP_{i}\left(u\right)\right)}\left(SP_{i}\left(u\right)\right)$ to perform calculation $x^{*}=y^{a_{i}b_{m}}PK{\left(P\left(u\right)\right)}^{\varpi }\;\mathrm{mod}\;N$.

Step 4: Storage server $STS_{l}$ determines whether equation $\varpi =H(i||m||\alpha_{i}||b_{m}||x*||SP_{i}\left(u\right))$ holds. If it is true, the signature authentication is correct, then go to Step 5; otherwise, go to Step 1.

Step 5: The storage server $STS_{l}$ receives the sensing data block $SP_{i}\left(u\right)$ and determines whether other sensing data blocks have been received. If yes, go to Step 6; otherwise, go to Step 1.

Step 6: The storage server $STS_{l}$ stores all the sensing data blocks $SP_{i}\left(u\right)\left|\right|\ldots \left|\right|SP_{h}\left(u\right)$ received by the server ($SP_{i}\left(u\right)\subset P\left(u\right)$, …, $SP_{h}\left(u\right)\subset P\left(u\right)$).

Step 7: End.

## 4. Security Analysis

The generation of keys, the overhead of key storage, the overhead of key renewal, and the signature and authentication of information are usually important indicators for measuring the transmission security and storage performance of blockchain-oriented sensor images. They are also important factors affecting the transmission and storage performance of inductive images. In this solution, the image sensing and sensing data blocks division algorithm of the smart image sensor, the public key generation algorithm of the sensing data blocks, the private key generation of the sensing data blocks and the update algorithm, the signature algorithm of the sensing data blocks, and the signature verification algorithm of the sensing data blocks are all important factors affecting the secure transmission and storage of the sensor image for the blockchain. Therefore, this section analyzes and proves the security aspect of this solution. We also analyze and prove the overhead aspects such as key storage overhead and key update overhead of this solution. The security function analysis table is shown in Table 2. The specific descriptions are as follows.

### 4.1. Security Analysis

#### 4.1.1. Resist Theft Attacks

This solution is the basic principle of security based on the RSA signature mechanism, that is, the difficulty of large composite prime factorization. In the IoT, it is often faced that a certain piece of image sensing data may be stolen in a certain period of time. How to ensure the security of image sensing data in the blockchain under this situation becomes very important. This solution can effectively defend against theft attacks of user sensor images by an attacker, and even if a certain piece of image sensing data is stolen, the remaining image sensing blocks cannot be stolen through this piece of data. The specific analysis is proved as follows.

**Theorem** **1.**
*In this solution, if the image sensing data block*
n=2
*, then the security of this solution is equal to the security of the Guillou–Quisquater (GQ) digital signature mechanism.*


**Proof of Theorem** **1.**When n=2, according to the public key generation algorithm of the sensing data blocks, the key parameter information $t_{1,1}={R_{1}}^{a_{2}}\;\mathrm{mod}\;N$ of the image sensing data block $SP_{1}\left(u\right)$ and the key parameter information $t_{2,1}={R_{1}}^{a_{1}}\;\mathrm{mod}\;N$ of the image sensing data block $SP_{2}\left(u\right)$ can be obtained. Even if an illegal user steals the key parameter information $t_{1,1}={R_{1}}^{a_{2}}\;\mathrm{mod}\;N$ of the image sensing data block $SP_{1}\left(u\right)$, if he wants to derive the key parameter information $t_{2,1}$ of the image sensing data block $SP_{2}\left(u\right)$, he will face the problem of solving the RSA problem. Similarly, even if an illegal user steals the key parameter information $t_{2,1}={R_{1}}^{a_{1}}\;\mathrm{mod}\;N$ of the image sensing data block $SP_{2}\left(u\right)$, if he wants to derive the key parameter information $t_{1,1}$ of the image sensing data block $SP_{1}\left(u\right)$, he will again face the problem of solving the RSA problem. □

**Theorem** **2.***When the number of image sensing data blocks is greater than 2, if*P(u)*is a user image collected by a smart image sensor, and*$SP^{\Lambda }\left(u\right)$*is a set of sensing image data blocks stolen by an illegal user, then*$P\left(u\right)-SP^{\Lambda }\left(u\right)$*is a data block set of sensing images whose sensing image data have not been stolen. Let*$SP_{i}\left(u\right)\subset P\left(u\right)$, $SP_{j}\left(u\right)\subset P\left(u\right)$, *and*$SP_{i}\left(u\right)\subset SP^{\Lambda }\left(u\right)$, $SP_{j}\left(u\right)⊄\left(P\left(u\right)-SP^{\Lambda }\left(u\right)\right)$. *The necessary and sufficient condition is: if and only if*$\gcd\left(MA/a_{i}:SP_{i}\left(u\right)\right)\subset SP^{\Lambda }\left(u\right))|\left(MA/a_{j}\right)$, *then the key parameter information*$t_{j,1}={R_{1}}^{MA/a_{j}}\;\mathrm{mod}\;N$*of the image sensing data block*$SP_{j}\left(u\right)$*can be derived from the key parameter information*$t_{i,1}={R_{1}}^{MA/a_{i}}\;\mathrm{mod}\;N$*of the image sensing data block*$SP_{i}\left(u\right)$.

**Proof of Theorem** **2.**(1) Let $\Delta =\gcd(MA/a_{i}:SP_{i}\left(u\right)\subset SP^{\Lambda }\left(u\right))$, then there exists a corresponding integer $e_{i}$ such that $L=\sum_{SP_{i}\left(u\right)\subset SP^{\Lambda }\left(u\right)}e_{i}*MA/a_{i}$. If $\gcd\left(MA/a_{i}:SP_{i}\left(u\right)\subset SP^{\Lambda }\left(u\right)\right)|\left(MA/a_{j}\right)$, then there is an integer *r* such that $MA/a_{j}=r*L$; then, $t_{j,1}={R_{1}}^{MA/a_{j}}\;\mathrm{mod}\;N={R_{1}}^{r*L}\;\mathrm{mod}\;N={R_{1}}^{r*\sum_{SP_{i}\left(u\right)\subset SP^{\Lambda }\left(u\right)}e_{i}*MA/a_{i}}\;\mathrm{mod}\;N=\prod_{SP_{i}\left(u\right)\subset SP^{\Lambda }\left(u\right)}{R_{1}}^{r*e_{i}*MA/a_{i}}=\prod_{SP_{i}\left(u\right)\subset SP^{\Lambda }\left(u\right)}{t_{i,1}}^{r*e_{i}}$. Therefore, $t_{j,1}$ can be derived from the key parameter information $t_{i,1}={R_{1}}^{MA/a_{i}}\;\mathrm{mod}\;N$ of the sensing image data block $SP_{i}\left(u\right)$.(2) If $t_{j,1}$ can be deduced from the key parameter information of all the sensing image data blocks in $SP_{i}\left(u\right)\subset SP^{\Lambda }\left(u\right)$, then according to the derivation in Part (1), it is easy to get $\gcd\left(MA/a_{i}:SP_{i}\left(u\right)\subset SP^{\Lambda }\left(u\right)\right)|\left(MA/a_{j}\right)$. □

**Theorem** **3.***In this solution,*$\gcd\left(MA/a_{i}:SP_{i}\left(u\right)\subset SP^{\Lambda }\left(u\right)\right)∤\left(MA/a_{j}\right)$.

**Proof of Theorem** **3.**Because $MA=a_{1}*a_{2}*\ldots *a_{n}=\prod_{i=1}^{n}a_{i}$, there is $a_{j}$|$\gcd(MA/a_{i}:SP_{i}\left(u\right)\subset SP^{\Lambda }\left(u\right))$. Because $\gcd(a_{i},a_{j})=1$ (i∈{1,2,3,…,n}, j∈{1,2,3,…,n}, i≠j), there is $a_{j}∤\left(MA/a_{j}\right)$, thus $\gcd\left(MA/a_{i}:SP_{i}\left(u\right)\subset SP^{\Lambda }\left(u\right)\right)∤\left(MA/a_{j}\right)$. □

In summary, from **Theorems 1**, **2**, and **3**, we can get that: In this solution, at each time period, it is difficult for illegal users to steal the image sensing data. According to the analysis of **Theorems 2** and **3**: In this solution, even if there are n−1 blocks of image sensing data in the image sensing data, and they are stolen in a certain period of time, it is still very difficult for illegal users to steal the remaining image sensing data blocks.

#### 4.1.2. Resist Replay Attack and Denial of Service (Dos) Attack

In the signature algorithm of the sensing data block proposed in this paper, the smart image sensor generates a timestamp value $TS_{i}$ of the smart image sensor after signing the sensing data block $SP_{i}\left(u\right)$, and then transmits the signature message and timestamp value to the storage server. In the signature verification algorithm of the sensing data block, when the storage server receives the signed message and the timestamp value $\left(\eta ||TS_{i}\right)$, another storage server timestamp value $TSTS_{i}$ is generated. The storage server verifies the freshness of the signed message by determining whether the session delay is within an allowable time interval Δt. Assume that the attacker obtains the signature message and timestamp of the verified sensing data block through illegal means, and attempts to destroy the correctness of the verification algorithm with the verified sensing data block. The storage server can ignore these duplicate signed messages by checking the freshness of the timestamp value, protect the correctness of the transmission and verification of the sensing data block, and effectively resist replay attacks. It can effectively reduce the consumption of network bandwidth by repeated information, reduce the occupation rate of the storage server algorithm when the repeated information is verified, and effectively resist denial of service attacks.

#### 4.1.3. Resist Counterfeit Attack and Server Camouflage Attack

In this solution, the smart image sensor generates different private keys for different sensing image data blocks to sign in each time period of the cycle *T*, and the private keys in different time periods are updated. Therefore, even if the attacker obtains the signature message $\eta =sign_{SK_{i,m}\left(SP_{i}\left(u\right)\right)}\left(SP_{i}\left(u\right)\right)$ of the sensing image data block $SP_{i}\left(u\right)$ stored in a certain storage server in a certain period by illegal means, and fakes the correct sensing image data block, it is still impossible to calculate $x^{*}=y^{a_{i}b_{m}}PK{\left(P\left(u\right)\right)}^{\varpi }\;\mathrm{mod}\;N$ through the storage server at other time periods, and the verification of the equation $\varpi =H(i||m||\alpha_{i}||b_{m}||x*||SP_{i}\left(u\right))$ is not successful. It can resist counterfeit attacks, protect the security of the sensing image data blocks, and avoid storing false or duplicate image block information, thereby effectively improving the storage efficiency of the storage server. Secondly, since the storage servers store the sensing image data blocks independently of each other and do not need to pass mutual authentication, a malicious attacker cannot disguise as a storage server to obtain the user sensing image data blocks stored in another storage server. It can resist server camouflage attack.

#### 4.1.4. Decentralization and Private Key Update

This solution uses the advantages of blockchain technology decentralization to intelligently divide the user’s sensory image data into *n* blocks, and the status between the data blocks is equal. Moreover, it is proved from the above **Theorems 2** and **3** that, even if a data block is illegally acquired or destroyed by an attacker, the remaining data block information cannot be obtained through this data block information, and the entire image sensing data cannot be affected. At each time period of the sensing cycle, the smart image sensor updates different private keys and generates different signatures for each image sensing data block, which undoubtedly increases the difficulty for illegal users to steal image sensing data, and enhances the security of image sensing data during transmission and storage.

### 4.2. Cost Analysis

In this solution, assume that the number of divided blocks of user image’s sensing data is *n*. At each time period in the cycle *T*, the key calculated by the smart image sensor is n+1, the key to be stored is *n*, and the key to be stored by the different storage servers in the blockchain is 1. During the entire cycle *T*, the total number of updates required by the smart image sensor is n∗T. The cost analysis results of this solution are shown in Table 3. The specific analysis and proof are as follows.

**Theorem** **4.**
*In this solution, when the image data are divided into n quickly in different time periods within the cycle T, the intelligent image sensor calculates the key is*
n+1
*, the key to be stored is n, and the key to be stored by different storage servers in the blockchain is 1.*


**Proof of Theorem** **4.**In this solution, $b_{i}$ represents the parameter generated in the *j*th time period in the cycle *T*, but *i* in $a_{i}$ is related to the number *n* of the divided blocks of the user’s image sensing data. According to the block division algorithm for image sensing and sensing data, the public key generation algorithm for sensing data blocks, and the private key generation and updating algorithm for sensing data blocks can determine the *j*th time period (*j* = 1,2, …, *T*) within the cycle *T*. The smart image sensor needs to calculate a unique private key for the signature transmission of each image sensing data block. In this way, since the image sensing data are divided into *n* blocks, *n* private keys need to be calculated. In the blockchain, only the unique public key is needed to authenticate the signatures of different sensing data blocks. Therefore, the unique public key is calculated in the public key generation algorithm of the sensing data block. Therefore, at different time periods in the cycle *T*, the key calculated by the smart image sensor is n+1, the key to be stored is *n*, and the key to be stored by different storage servers in the blockchain is 1. □

**Theorem** **5.***In this solution, if the image data are divided into n quicklu, in the jth time period within the cycle T, the total number of keys that the smart image sensor has updated is*n∗(j−1).

**Proof of Theorem** **5.**(1) During the j=1 time period in the cycle *T*, the intelligent image sensor can only generate a privacy key for each image sensing data block, and there is no key update process. Therefore, the total number of keys updated by the system is 0, thus the conclusion holds.(2) Assume that the conclusion is valid during the j=i time period (*j* = 1,2, …, *T*) in the cycle *T*, that is, the total number of keys that the smart image sensor has updated is n∗(i−1). When j=i+1, during the transition of the intelligent image sensor from the *i*th time period to the i+1 time period, each image sensing data block needs to update its own private key. Therefore, during the transition, the total number of key updates is *n*. Since the total number of keys updated by the smart image sensor during the ith period is n∗(i−1), the total number of keys already updated by the smart image sensor during the i+1 period is n∗(i−1)+n=n∗i. Thus, the conclusion holds. □

## 5. Conclusions and Future Work

This paper proposes a security transmission and storage solution about sensing image for blockchain in the IoT. This solution mainly includes five algorithms: the blocking algorithm for image sensing and data sensing of smart image sensors; the public key generation algorithm for sensing data block; the private key generation and update algorithm for sensing data block; the signature algorithm for sensing data block; and the signature verification algorithm for sensing data block. To prove that the solution can safely transmit and store the user’s sensory image information, we conducted a series of security analysis. The proof results show that the solution can resist attacks such as theft and forgery, and ensure the security of user image information during transmission and storage.

In future work, we intend to apply digital image encryption technology to user-induced image encryption. Multi-encryption is used to ensure that user images are not stolen by attackers. Even if they are obtained by others, no valuable information can be obtained from the image information. Therefore, the technology can effectively prevent attackers from forging user image information to deceive authentication, and enhance the security of secure image transmission and storage solution.

## Figures and Tables

**Figure 1 sensors-20-00916-f001:**
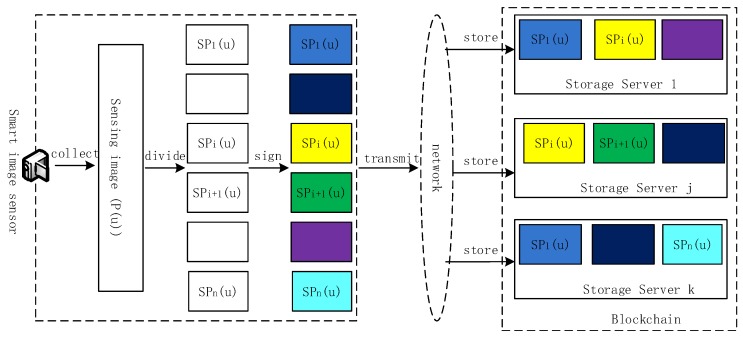
Schematic diagram of secure transmission and storage of blockchain-oriented sensory images in the IoT.

**Table 1 sensors-20-00916-t001:** Symbols and definitions.

Symbol	Definition
*u*	User *u*
P(u)	Images of user *u*
$SP_{i}\left(u\right)$	*i*th block in user image
Φ	Empty
*A*⊂*B*	*A* is included in *B*
∩	Union operation
∪	Intersection
||	String operation
*A*|*B*	*A* divisible by *B*
*A* ∤ *B*	*A* is not divisible by *B*
*T*	Cycle *T* of smart image sensor
PK(P(u))	Public key of user image sensing data P(u)
$SK_{i,m}\left(SP_{i}\left(u\right)\right)$	Private key of sensing data block $SP_{i}\left(u\right)$ in time period t=m
$STS_{l}$	Storage server *l*
$TS_{i}$	A timestamp value generated by the smart image sensor
$TSTS_{i}$	A timestamp value generated by the storage server
$sign_{SK(A)}\left(B\right)$	Data *B* is signed with the private key SK(A)
$SP^{\Lambda }\left(u\right)$	Set of sensory image data blocks stolen by illegal users

**Table 2 sensors-20-00916-t002:** Analysis of safety functions.

Security Function	Our Solution
Resist theft attacks	√
Resist replay attack	√
Resist denial of service (DOS) attack	√
Resist counterfeit attack	√
Resist server camouflage attack	√
Decentralization	√
Private key update	√

**Table 3 sensors-20-00916-t003:** Solution cost analysis.

Period	Number of Keys Calculated by Smart Sensor	Number of Keys Updated by Smart Sensor	Number of Keys Stored per Storage Server	Total Number of Keys Updated by Smart Sensor
1st time period	n+1	*n*	1	0
*j*th time period	n+1	*n*	1	n∗(j−1)
The entire cycle *T*	n∗T+1	n∗T	*T*	n∗T

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
