# Peer review of "A Security Transmission and Storage Solution about Sensing Image for Blockchain in the Internet of Things"

_sensors, 2020, doi:10.3390/s20030916_

Round 1

Reviewer 1 Report

Thanks for your effort, I made number of comments and suggestions in the attached file to be considered. 

Good luck

Author Response

Thank you very much for your comments on our article. All comments have greatly helped our article. Our response to these comments is as follows.

Reviewer 2 Report

This paper proposes a security transmission and storage solution about sensing image for blockchain in the Internet of Things, as claimed by the authors. On the whole, I think the authors have addressed an important topic and the overall work is substantive. I have a few comments for the purpose of the improvement:

1. There are only three keywords. Two more are required.
2. The statement of the main contribution is very weak. Contributions shall be an independent work that is different from other work or prior achievements, rather than a list of done work. For example, the statement of "The blockchain technology is combined with the IoT to store data for the IoT" does not make much sense when considering the contribution.
3. The structure of the related work section shall be improved. Listing other people's work generally is not a good way to describe research background. Comparisons are needed. Moreover, reviewed references shall be grouped into a number of topics.
4. Important references are missing. Suggest the authors review and cite following work in the field of blockchain in order to strengthen the reference: "Privacy-preserving Energy Trading Using Consortium Blockchain in Smart Grid", "Controllable and trustworthy blockchain-based cloud data management", "Permissioned Blockchain and Edge Computing Empowered Privacy-preserving Smart Grid Networks".
5. Experiment evaluations are needed.

Author Response

(The authors gave the same response as above.)

Round 2

Reviewer 1 Report

Thanks for applying most of the suggestions and revising the paper according to the comments.